# The PROVIT Study—Effects of Multispecies Probiotic Add-on Treatment on Metabolomics in Major Depressive Disorder—A Randomized, Placebo-Controlled Trial

**DOI:** 10.3390/metabo12080770

**Published:** 2022-08-21

**Authors:** Kathrin Kreuzer, Alexandra Reiter, Anna Maria Birkl-Töglhofer, Nina Dalkner, Sabrina Mörkl, Marco Mairinger, Eva Fleischmann, Frederike Fellendorf, Martina Platzer, Melanie Lenger, Tanja Färber, Matthias Seidl, Armin Birner, Robert Queissner, Lilli-Marie Stefanie Mendel, Alexander Maget, Alexandra Kohlhammer-Dohr, Alfred Häussl, Jolana Wagner-Skacel, Helmut Schöggl, Daniela Amberger-Otti, Annemarie Painold, Theresa Lahousen-Luxenberger, Brigitta Leitner-Afschar, Johannes Haybaeck, Hansjörg Habisch, Tobias Madl, Eva Reininghaus, Susanne Bengesser

**Affiliations:** 1Psychosomatics and Psychotherapy Clinical Department of Psychiatry and Psychotherapeutic Medicine, University Hospital for Psychiatry, Medical University of Graz, 8036 Graz, Austria; 2Neuropathology and Molecular Pathology, Institute for Pathology, Medical University of Innsbruck, 6020 Innsbruck, Austria; 3Diagnostic and Research Center for Molecular Biomedicine, Institute of Pathology, Medical University of Graz, 8010 Graz, Austria; 4Institute for Psychology, Otto Friedrich University of Bamberg, 96047 Bamberg, Germany; 5Research Unit Integrative Structural Biology, Division for Molecular Biology and Biochemistry, Gottfried Schatz Research Center, Medical University of Graz, 8036 Graz, Austria; 6BioTechMed Graz, 8010 Graz, Austria

**Keywords:** depression, gut–brain-axis, probiotics, metabolomics, NMR spectroscopy, butyrate, randomized controlled trial

## Abstract

The gut–brain axis plays a role in major depressive disorder (MDD). Gut-bacterial metabolites are suspected to reduce low-grade inflammation and influence brain function. Nevertheless, randomized, placebo-controlled probiotic intervention studies investigating metabolomic changes in patients with MDD are scarce. The PROVIT study (registered at clinicaltrials.com NCT03300440) aims to close this scientific gap. PROVIT was conducted as a randomized, single-center, double-blind, placebo-controlled multispecies probiotic intervention study in individuals with MDD (*n* = 57). In addition to clinical assessments, metabolomics analyses (1H Nuclear Magnetic Resonance Spectroscopy) of stool and serum, and microbiome analyses (16S rRNA sequencing) were performed. After 4 weeks of probiotic add-on therapy, no significant changes in serum samples were observed, whereas the probiotic groups’ (*n* = 28) stool metabolome shifted towards significantly higher concentrations of butyrate, alanine, valine, isoleucine, sarcosine, methylamine, and lysine. Gallic acid was significantly decreased in the probiotic group. In contrast, and as expected, no significant changes resulted in the stool metabolome of the placebo group. Strong correlations between bacterial species and significantly altered stool metabolites were obtained. In summary, the treatment with multispecies probiotics affects the stool metabolomic profile in patients with MDD, which sets the foundation for further elucidation of the mechanistic impact of probiotics on depression.

## 1. Introduction

Major depressive disorder (MDD) affected 300 million people worldwide in 2018, with an individual’s lifetime prevalence ranging from 16–20%, [1]. During the COVID-19 pandemic, the prevalence even increased to 27.6%, with an additional 53.2 million cases of MDD globally [2]. The symptoms of depression cause tremendous individual suffering, including depressed mood, lack of energy, anhedonia, disturbed sleep, changes in appetite, cognitive deficits, and in its severest form, suicidality [3].

The complex pathogenesis of MDD involves concatenation of polygenic inheritance, chronic stress, as well as acute triggers. MDD elicits a high genetic burden with concordance rates of 50% for monozygotic twins and even 15–20% for dizygotic twins [4]. Recently, the latest meta-analysis from the Psychiatric Genomics Consortium (PGC) MDD working group identified over one hundred predisposing, genome-wide significant gene variants [5]. Nevertheless, the polygenic orchestra of genes on its own is not sufficient to trigger the onset of a depressive episode [6]. From a molecular perspective, gene–environment interactions, such as disturbed circadian rhythms, neurotransmitter imbalance, chronic low-grade inflammation, oxidative stress, and the gut–brain axis, mediate the outbreak of depressive episodes [5,7,8,9,10,11,12]. In addition, psychosocial factors, such as early childhood trauma, the lack of coping mechanisms, and traumatic life events, play a significant role in the complex pathophysiology of depression [13].

Notably, the microbial host environment in the gut contributes to the core domains of affective disorders [14]. Recent meta-analyses exhibited profound gut microbiome disruptions in patients suffering from depression [15,16]. One of the largest projects to date, the Flemish gut flora project, discovered reduced levels of butyrate-producing species *Coprococcus* spp. and *Dialister* in patients with MDD. The relative abundance of butyrate-producing *Faecalibacterium* and *Coprococcus* correlated positively with RAND scores, which are quality of life indicators, as assessed with the RAND-36 health-related quality of life survey [15]. Another meta-analysis by Sanada et al. (2020), which included ten observational (*n* = 701) and six interventional trials (*n* = 302), revealed that patients with MDD overall show a decreased abundance of the bacteria family *Prevotellaceae*, and the genera *Corprococcus* and *Faecalibacterium*, compared to non-depressed individuals [16].

According to the current literature, metabolites released from the gut microbiome can affect the pathophysiological networks of the host [17]. In general, gut bacteria produce short chain fatty acids (SCFAs), such as acetate, propionate, and butyrate, via the fermentation of dietary fiber. In addition to their anti-inflammatory effect and their influence on the human metabolism, SCFAs have been shown to have beneficial effects on psychiatric symptoms, as they can pass through the blood–brain barrier [18,19].

Individuals suffering from MDD reveal a distinctly different metabolome. The largest metabolomics meta-analysis, which included 5283 study participants with depression and 10,145 healthy controls, identified 230 metabolic markers, with 51 metabolic markers distinguishing the groups and remaining significant after adjusting for smoking, age, sex, fasting status, and lipid-modifying drugs [20]. A further recent meta-analysis by Konjevod et al. (2021) described a potential interconnection between neurodegenerative and neuropsychiatric disorders and observed overall alterations in the microbiome metabolism [21].

Especially for MDD, metabolomics (NMR) analyses of randomized, double-blind, placebo-controlled multispecies probiotic trials are still lacking. The present PROVIT study constitutes the first study worldwide in MDD that analyzed microbiome, metabolome, gene expression, and cognition changes after a randomized, double-blind, placebo-controlled, multispecies probiotic intake with state-of-the-art molecular biological methods [22,23]. In order to analyze the interconnection between decreasing inflammation and the shifting microbial composition to relatively more abundant butyrate-producing bacteria in the probiotics group of the PROVIT study, we used NMR metabolomics as the appropriate tool to find a potential missing link. Hence, NMR simultaneously measures all metabolites present in a given sample (e.g., serum, urine, stool) [24].

Our in-depth characterization of the PROVIT study samples presented here broadens the knowledge about the effects of probiotics add-on therapy in patients with MDD by further investigating the gut microbiome as well as the metabolomic profile of serum and stool. Thus, the (I) major aim includes the investigation of the molecular effects of multispecies probiotics on the metabolomic profile of serum and stool in study participants suffering from MDD. More precisely, we analyzed (a) changes in the metabolome over time of the placebo and probiotic group (time effect) and (b) alterations in the metabolome between the groups taking placebo vs. probiotics (group effect). In addition, our second goal (II) was to study the relationship between the gut microbiome and significantly altered metabolites in univariate analyses.

## 2. Materials and Methods

### 2.1. Study Design

The PROVIT trial was conducted as a monocentric, double-blind, randomized, placebo-controlled, multispecies probiotic intervention study. The participants were recruited at the Clinical Department of Psychiatry and Psychotherapeutic Medicine at the Medical University of Graz by psychiatrists and psychologists of the research unit for affective disorders managed by Univ.Prof. Eva Reininghaus. PROVIT aimed to investigate the effects of probiotic add-on treatment (OMNi-BiOTiC^®^ Stress Repair by Allergosan) on affective symptoms, cognitive function, microbiome, metabolome, routine blood markers, and targeted gene expression. The study procedure of the PROVIT study was described in detail in the publications by Reininghaus et al. (2020) [25] and Reiter et al. (2020) [23]. Inpatients with MDD received treatment as usual and a multispecies-probiotic/vitamin B7 or a placebo/vitamin B7 drink as an add-on therapy for 28 days. Stool samples were collected at three points in time: Before (t_0_), after two weeks (t_1_), and at the end of the intervention (t_2_). Serum samples were gathered at two points in time: Before (t_0_) and at the end of the intervention (t_2_) of the PROVIT study. An overview of the PROVIT study design is illustrated in Figure 1.

The local ethics committee approved the study (EK 29-235 ex 16/17), and the randomized controlled trial (RCT) was registered at clinicaltrials.com (NCT03300440). All patients provided written informed consent. At the time point of inclusion, demographic and clinical parameters (age, weight, height, body mass index (BMI), sex, and medication), cognitive tests and lifestyle questionnaires were assessed. The diagnosis of MDD was verified by an experienced psychiatrist according to the International Statistical Classification of Diseases and Related Health Problems (ICD-10) guidelines and the Mini International Neuropsychiatric Interview (M.I.N.I) [26]. The severity of depressive symptoms was measured with the Hamilton Rating Scale for Depression (HAMD) [27] and the Beck Depression Inventory-II (BDI-II) [28]. The exclusion and inclusion criteria, as well as the cognitive and psychological inventories, are described in Reininghaus et al. 2020 [25] and Reiter et al. 2020 [23].

### 2.2. Details of the Probiotic Supplementation

Individuals subjected to the verum group received a multistrain probiotic ‘OMNi-BiOTiC^®^ Stress Repair’, which was provided by the ‘Institute Allergosan’ and produced by Winclove BV, Amsterdam, Netherlands. ‘OMNi-BiOTiC^®^ Stress Repair (SR)’ is commercially available and includes nine bacterial strains with ≥2.5 × 10^9^ colony-forming units (CFU) per gram. One probiotic drink contained at least 3 g, which sums up to a total number of 7.5 × 10^9^ CFU per bag. More precisely, one multispecies probiotic drink included the following bacterial strains: *Bifidobacteria* (*B. bifidum W23, B. lactis W51, B. lactis W52*) and *Lactobacilli* (*L. acidophilus W22, L. casei W56, L. paracasei W20, L. plantarum W62, L. salivarius W24, L. lactis W19*). Further ingredients in the probiotic and placebo product included D-biotin (Vitamin B7), common horsetail, fish collagen, and keratin. The matrix included maize starch, maltodextrin, inulin, potassium chloride, magnesium sulfate, fructooligosaccharides (FOS), enzymes (amylases), and manganese sulfate. The drink was provided by a psychiatrist on call from the research unit every day in the morning from 7 to 8 o’clock before breakfast. The doctors on call were in charge of stirring the drink in a double-blind setting and surveyed the intake of the probiotic/placebo drink.

Individuals subjected to the placebo group received only a similar-looking placebo drink containing the matrix and D-biotin (vitamin B7), which was added to both formulas due to considerations of the ethics committee that required both groups to receive a beneficial substance that does not directly influence the microbiome. To be even more precise, ‘OMNi-BiOTiC^®^ SR’ contains probiotics and, to a minor degree, prebiotics, which can be declared as synbiotic. For readability, we will refer to ‘OMNi-BiOTiC^®^ SR’ as a probiotic. The placebo product had the same color, consistency, and taste as the probiotic product.

### 2.3. Metabolomics

#### 2.3.1. Reagents

Dibasic sodium phosphate (Na_2_HPO_4_), sodium hydroxide, hydrochloric acid (32% *m*/*v*), and sodium azide (NaN_3_) were obtained from VWR International (Darmstadt, Germany). From Alfa Aesar (Karlsruhe, Germany), 3(trimethylsilyl) propionic acid-2,2,3,3-d_4_ sodium salt (TSP) was obtained. Deuterium oxide (D_2_O) was obtained from Cambridge Isotopes laboratories (Tewksbury, MA, USA). Deionized water was purified using the in-house Milli-Q Advantage Water Purification System from Millipore (Schwalbach, Germany). All chemicals were used with no further purification. The phosphate NMR buffer solution was prepared by dissolving 5.56 g of anhydrous Na_2_HPO_4_, 0.4 g of TSP, and 0.2 g NaN_3_ in 400 mL of D_2_O and adjusted to pH 7.4 with 1M NaOH and HCl. Upon addition of D_2_O to a final volume of 500 mL, the pH was re-adjusted to pH 7.4 with 1M NaOH and HCl.

#### 2.3.2. Metabolomic Quantification Using NMR

A methanol–water solution was added in a 2:1 ratio to stool and serum samples (200 µL of serum plus 400 µL of methanol; 1 g of stool plus 2 mL of methanol) to remove proteins and to quench enzymatic reactions. Then, samples were lysed using the precellys homogenizer and stored at −20 °C for one hour until further processing. In the next step, the samples were spun at 17,949 rcf at 4 °C for 30 min. The supernatants were lyophilized, and 500 µL of NMR buffer in D_2_O was added to the samples, which were then transferred to 5 mm NMR tubes. All NMR experiments were performed at 310 K on an AVANCE™ Neo Bruker Ultrashield 600 MHz spectrometer equipped with a TXI probe head and processed as described previously [29]. Shortly, the 1D CPMG (Carr-Purcell-Meiboom-Gill) pulse sequence (cpmgpr1d, 512 scans, 73,728 points in F1, 11904.76 HZ spectral width, 512 transients, recycle delays 4 s) with water suppression by pre-saturation was used for ^1^H 1D NMR experiments. Bruker Topspin version 4.0.2 was used for NMR data acquisition. Spectra for all samples were automatically processed (exponential line broadening of 0.3 Hz), phased and referenced using TSP at 0.0 ppm, using the Bruker Topspin 4.0.2 software (Bruker GmbH, Rheinstetten, Germany). Spectra were imported to Matlab2014b, and the regions around the water, TSP, and remaining methanol signals were excluded. For correcting the sample metabolite dilution, a probabilistic quotient normalization was performed [30]. Data analysis and the preparation of figures were carried out in MetaboAnalyst 5.0 (*vide infra*).

### 2.4. Microbiome Analysis

For microbiome analysis, 1 g of the stool sample was collected from each participant at all three timepoints. Samples were immediately stored in a −80 °C freezer until sequence analysis, which was performed according to the supplier’s recommendations with Illumina MiSeq [31]. DNA was extracted according to the manufacturer’s instructions with Magna Pure LC DNA III Isolation Kit (Bacteria, Fungi; Roche, Mannheim, Germany). Polymerase chain reaction (PCR) was used to amplify V3–V4 regions of the bacterial 16S rRNA gene from fecal total DNA with the following target-specific primers: MyOv3v4F—CCTACGGGNGGCWGCAG and MyOv3v4R—GACTACHVGGGTATCTAATCC. PCR amplification products were pooled, indexed, and purified as described by [31]. An Illumina MiSeq desktop sequencer with v3 chemistry and 600 cycles (2 × 300) was used to sequence the final library. FASTQ files were used for data analysis. Obtained reads were processed using Quantitative Insights Into Microbial Ecology (QIIME, v1.9.1) scripts on the galaxy server of the Medical University of Graz (galaxy.medunigraz.at). Paired-end reads were pre-filtered, trimmed, and filtered for quality and chimeras using the DADA2 library. DADA2 was used to assign a taxonomy against the SILVA SSURef database (release v132). The RDP classifier was used for the taxonomic assignment using default parameters. A biom table was constructed for downstream analyses. The data for this study were deposited in the European Nucleotide Archive (ENA) at EMBL-EBI under accession number PRJEB40986 (http://www.ebi.ac.uk/ena/data/view/PRJEB40986, accessed on 24 July 2022). The microbiome analysis and bioinformatic analysis were conducted as previously published by Reininghaus et al. [25].

### 2.5. Statistics

To identify changes in the metabolomics profiles between the probiotics and placebo group (group effect) and before and after the 28-day intervention (time effect), multivariate statistical analysis was performed as described previously [32]. NMR data were analyzed in Matlab^®^ vR2014a (Mathworks, Natick, MA, USA), using principal component analysis (PCA), sparse partial least squares-discriminant analysis (sPLS-DA), and orthogonal-partial least squares-discriminant analysis (O-PLS-DA) [33]. The statistical significance was validated using associated data consistency checks and 7-fold cross-validation, expressed by Q^2^. Figures of PCA, sPLS-DA, OPLS-DA plots, and volcano plots were prepared using a web-based version of MetaboAnalyst 5.0 [29]. Biomarker identification based on the area under the receiver operator characteristics curve (AUROC) was performed to further discriminate the groups. Metabolites with the lowest *p*-values were included in the group comparison tests. Data were represented as medians (minimum-maximum). The *p*-values were calculated using a two-tailed Student’s *t*-test for pairwise comparison of variables. To analyze the time x group interaction, a two-way ANOVA was used. The correlations between NMR-based metabolomics data and 16S rRNA sequencing data over the PROVIT metabolomics cohort were calculated using Spearman’s rank-order correlation coefficient. All other statistical analyses were performed with SPSS 26 (SPSS Inc., Chicago, IL, USA). Box plot diagrams were illustrated with GraphPad Prism 8.0 (GraphPad Software, La Jolla, CA, USA). The cohort description is depicted in Figure 1. The groups did not differ in age, sex, and medication; therefore, it was not necessary to correct for these variables in the major analyses.

For descriptive statistics, in order to test for associations between the two groups, Student *t*-tests, Mann-Whitney-U-tests, or Chi-square tests were used, depending on the normal distribution of data as assessed with Shapiro-Wilk tests. The level of significance was set to *p* < 0.05.

## 3. Results

### 3.1. Descriptive Statistics of the PROVIT Metabolomics Cohort

A detailed description of the PROVIT sample was already published by Reininghaus et al. [25]. In total, 28 individuals from the intervention group and 29 individuals from the placebo group provided complete stool samples for metabolomics analysis. At baseline, there were no significant differences between the two groups in age, sex, smoking, dairy products before the trial, BMI, waist-to-hip ratio (WHR), years of education, illness duration in years, and hemoglobin A1c (HbA1c), as seen in Table 1. The psychopharmacological treatment of the cohort is presented in Table 2. Six individuals in the intervention group and two individuals in the placebo group were not pre-medicated. Fourteen and nineteen participants took medication from one or two different substance categories, and eight and twelve individuals took drugs involving three or more distinct substances.

The mean values and standard deviation of depression scores of the PROVIT metabolomics cohort are shown in Table 3. The results of the repeated-measures ANOVA with the two independent factors time point (beginning and end of the study) and group (intervention vs. placebo) and dependent variables BDI-II and HAMD revealed that both groups significantly improved in depression scores over time (BDI-II: *F_(_*_1,52)_ = 103.38, *p* < 0.001, *η^2^* = 0.67; HAMD: *F*_(1,54)_ = 42.17, *p* < 0.001, *η^2^* = 0.44; see Table 2). There was no significant group effect (BDI-II: *F*_(1,52)_ = 2.00, *p* = 0.162, *η^2^* = 0.04; HAMD: *F*_(1,52)_ = 0.56, *p* = 0.459, *η^2^* = 0.01), nor a significant time-group-interaction (BDI-II: *F*_(1,52)_ = 0.29, *p* = 0.592, *η^2^* = 0.01; HAMD: *F*_(1,52)_ = 0.02, *p* = 0.879, *η^2^* = 0.00). For detailed results of the PROVIT cohort on psychiatric scales, see Reininghaus et al. (2020) [25].

### 3.2. Untargeted Metabolomic Assessment of Serum Reveals No Significant Differences

In the placebo group, O-PLS-DA comparisons revealed acceptable clustering, with cross-validation scores Q^2^ of 0.0787 and a *p*-value of 0.02, as seen in Figure 2A. In the probiotics group, O-PLS-DA comparisons showed non-acceptable clustering, with cross-validation scores Q^2^ of 0.0787 and a *p*-value of 0.77, as seen in Figure 2B. The model represented a correlation coefficient of R2Y = 0.52 and *p* = 0.45.

### 3.3. NMR-Based Metabolomics Analysis of Stool Reveals Changes Induced by Probiotics

In order to characterize changes in stool metabolic phenotypes upon treatment with probiotics, we carried out an NMR-based analysis of the available stool samples. PCA analysis of the six groups revealed clustering of most samples with principal component (PC) 1 of 89.4% and PC 2 of 9.1%—see Figure 3A. The plot shows strong variations of metabolite concentrations, with PC1 being dominated by acetate and PC2 being dominated by succinate, as seen in Appendix A. sPLS-DA analysis was used to identify differences between the study groups and the most predictive features. The sPLS-DA score plot of all six groups (placebo (*n* = 29) vs. probiotics (*n* = 28) at time points t_0_, t_1_, and t_2_) revealed a strong overlap of the metabolic profiles of all six groups, with Component 1 of 18.9% and Component 2 of 11.6%, as seen in Figure 3B. To obtain a better overview of the metabolic changes underwent in the placebo and probiotics groups, respectively, we carried out separate analyses of the two groups. The sPLS-DA score plot of all three subgroups of the placebo group (placebo at time points t_0_, t_1_, and t_2_) revealed no distinctive differences with Component 1 of 3.7% and Component 2 of 14.3%, as seen in Figure 4B.

In contrast, the probiotic intervention group showed a visible shift in the metabolic profiles over time (probiotics at time points t_0_, t_1_, and t_2_), as displayed by the sPLS-DA score plot, with Component 1 of 19.9% and Component 2 of 9.1%, as seen in Figure 4D. Interestingly, the metabolic profile seemed to become more heterogeneous during the probiotic treatment, an effect that we did not observe in the placebo group, as shown in Figure 4A,C. In order to identify the metabolites causing the changes observed in the sPLS-DA analysis, we carried out pairwise comparisons of the groups. While we could not observe any significant changes in the placebo group (*n* = 29) over the duration of the intervention (see Appendix A), in line with the sPLS-DA analysis, the concentrations of several metabolites changed significantly in the probiotic group (*n* = 28), as seen in Figure 5 and Figure 6.

Comparing the start and endpoints of the study, concentrations of trimethylamine and capric acid increased after the period of 28 days of multispecies probiotic supplementation, as seen in the volcano plot in Figure 5. Significant changes were observed in the probiotics group compared to the placebo after 28 days, as seen in Figure 6. At admission, we could detect slight differences between the two groups (Appendix A). After 28 days, these differences disappeared and down/upregulation of a set of nine metabolites was observed, as seen in Figure 6. Among those, we found increased levels of branched-chain amino acids (BCAAs) valine and isoleucine and the amino acids lysine and alanine, as seen in the volcano plot in Figure 6. Moreover, we found increased levels of methylamine, sarcosine, and the SCFA butyric acid after four weeks of multispecies probiotic intervention. Gallic acid, a polyphenol metabolite, decreased after four weeks of probiotic add-on treatment, as presented in the volcano plot in Figure 6. The Appendix A shows the most pronounced changes in stool metabolites illustrated in boxplots, as shown in Appendix A. Normalized concentrations of butyric acid, lysine, alanine, sarcosine, isoleucine, methylamine, and valine were significantly increased in the probiotics group (*n* = 28) in comparison to the placebo group (*n* = 29), as listed in Appendix A. In receiver-operating characteristic (ROC) analysis, seven ratios of significantly changed metabolites reached an area under the curve of 0.7, indicating that these could be used to assess the impact of probiotics on the stool metabolome. No statistical differences of significantly altered metabolites, calculated with Student’s *t*-test in the additional univariate analysis with ANOVA, were obtained.

### 3.4. Strong Correlations between Metabolites and Microbial Species Exist

As microbiota metabolize, synthesize, and release metabolites, the interconnection between microbiota species and metabolites is of major interest. We observed strong correlations between the four significantly altered metabolites (valine, butyric acid, capric acid, and isoleucine) in the probiotics group, with other metabolites and a relative abundance of bacterial species—for details, see Figure 7. For example, *Faecalibacterium* correlated positively with butyrate. Similarly, isoleucine positively correlated with *Ruminococcaceae* (see Figure 7). Strong correlations between the other five significantly altered metabolites (sarcosine, alanine, lysine, methylamine, and gallic acid) in the probiotics group, with other metabolites and the relative abundance of bacterial species, were observed and are represented in Appendix A.

## 4. Discussion

The current PROVIT investigation, which was recruited at the Department of Psychiatry and Psychotherapeutic Medicine at the Medical University of Graz in Austria, showed a significant shift in the stool metabolomic profiles of depressed individuals receiving 28 days of multispecies probiotic treatment provided by Allergosan. The stool metabolomic phenotype in the probiotics group shifted towards significantly higher normalized concentrations of butyrate, alanine, valine, isoleucine, sarcosine, methylamine, and lysine. Gallic acid was significantly decreased in the probiotic intervention group after four weeks of probiotic formula intake in the PROVIT study. In contrast, no significant metabolomic changes were observed in the placebo group. Similarly, the metabolomic pattern in serum showed no significant differences after the PROVIT add-on treatment in both groups. No statistical differences between the probiotics and placebo groups in depression scores could be obtained.

As MDD evidently shows alterations in the gut microbiome and metabolome, probiotics are discussed as potential powerful add-on therapy in the treatment of depression [34,35]. So far, only a few metabolomics studies in somatic disorders have analyzed changes in the metabolic profile of serum and stool after probiotics or prebiotics intervention in randomized controlled trials (RCTs). Metabolomics analyses after placebo-controlled probiotic intervention studies are scarce in the broad field of psychiatry. One prospective randomized, double-blind, placebo-controlled study conducted by Pinto-Sanchez et al. (2018) revealed that a six-week intervention with *Bifidobacterium longum NCC3001* (BL) led to reduced anxiety and depression scores in patients suffering from irritable bowel syndrome (IBS; *n* = 22) compared to a placebo control group (*n* = 22). In individuals with IBS receiving probiotic treatment, reduced amygdala and frontal-limbic activity levels were displayed in functional magnetic resonance imaging (fMRI). Additionally, patients with IBS showed a different metabolite profile in the probiotic group with decreased levels of phenylacetylglutamine, trimethyl-N-oxide, and 4-cresol in ^1^H NMR metabolomics [36].

The current analysis of the PROVIT trial reveals new aspects related to the metabolic impact of the probiotic intervention in MDD and extends our previous work by Reiter et al. (2020), in which we discovered decreasing levels of *IL-6* gene expression in the probiotics group. In contrast, *IL-6* gene expression levels increased in the placebo group [23]. Remarkably, butyrate and gallic acid have already been shown to be anti-inflammatory in vitro and in vivo in several studies [37]. Reininghaus et al. (2020) observed previously that probiotic supplementation in the PROVIT study resulted in a higher relative abundance of two butyrate-producing bacterial species in the gut microbiome, namely *Coproccocus 3* and *Ruminoccocus grauvanii* [25]. The subsequent changes in the PROVIT trial in normalized concentrations of butyrate and gallic acid in the stool’s metabolic phenotype in individuals receiving probiotic add-on therapy are a potential missing link between the probiotic treatment-induced alteration of the bacterial composition and the decreasing *IL-6* gene expression levels. Additionally, the probiotic intake obviously favors the growth of butyrate fermenters [25]. From a more general perspective, butyrate is an essential microbial metabolite in the colon, hence it nourishes colonocytes and strengthens their intercellular integrity [38,39,40]. As butyrate has favorable effects on sleep structure, probiotics that positively affect the existing butyrate fermenters may be used in treating sleeping disorders, which are very common in MDD [41]. Aside from the important effects of butyrate on sleep, butyrate plays a role in immunomodulation and differentiation of colonic regulatory T cells [42,43]. Furusawa et al. observed that luminal concentrations of butyrate and other SCFAs strongly correlate with the presence of regulatory T cells in the colon [43]. The example of butyrate underlines the important role of microbial-derived metabolites in Psychiatry. However, not only do SCFAs seem to play a vital role in the gut–brain axis, but amino acids have also been shown to be key players in the pathogenesis of depression [44,45,46].

Interestingly, 28 days of probiotic intake led to significantly elevated levels of the essential amino acids valine, isoleucine, alanine, and lysine in the PROVIT study. Sarcosine or N-methylglycine, another metabolite of the amino acid metabolism, was significantly increased in individuals receiving the probiotic add-on treatment. The largest metabolomics meta-analysis to date identified 230 metabolites and discovered that higher levels of isoleucine and tyrosine were correlated with increased odds of depression [20]. Previous research focusing on rat models revealed that depressed rats show decreased levels of amino acids in their fecal metabolome, compared to their respective healthy controls [20]. Another metabolomics study in patients with MDD discovered that individuals suffering from depression elicit a differential fecal metabolomic signature with an altered amino acid metabolism [47]. To conclude, probiotic treatment in the PROVIT trial potentially counteracted the divergent metabolomic fecal profile in patients with depression [47].

Bacteria in the gut play a major role in the macronutrient catabolism of proteins, carbohydrates, and lipids [48]. *Clostridia*, for example, exhibit a mechanism of amino acid catabolism—the Stickland reaction, which involves the coupled oxidation and reduction of two amino acids [49]. The PROVIT trial revealed strong correlations between amino acids and bacterial species after four weeks of the probiotic intervention. Remarkably, *Coprococcus 3* and *Ruminococcus grauvanii*, belonging to the bacterial family of *Clostridia*, were relatively more abundant. Tying this together, Stickland reactions may have been involved in the alteration of the metabolomic phenotype and created a potentially favorable surrounding for butyrate-producing species, which led to potentially anti-inflammatory diversity in the PROVIT study.

Another metabolite that elicits anti-inflammatory properties is gallic acid, which was also significantly altered after four weeks of probiotic treatment [50,51]. Gallic acid is a product of polyphenol metabolism and belongs to the subgroup of non-flavonoids [52]. In general, gallic acid is absorbed quickly by the colonic cells, and only the unabsorbed fraction is excreted with feces. Individuals receiving probiotic treatment in the PROVIT trial showed significantly reduced gallic acid levels. The reduction of the metabolite gallic acid can be interpreted in two ways. Either probiotic bacterial strains might have shifted to a bacterial composition with a higher abundance of bacterial species capable of producing gallic acid, which led to increased uptake by enterocytes and thereby resulted in a decreased abundance of gallic acid in stool. The other possible interpretation is that reduced levels of gallic acid are a probable direct result of multispecies probiotic treatment [53]. Specifically, *Lactobacillus plantarum*, a bacterial species included in the probiotic drink in the PROVIT study, has the bio-transformational capability to degrade gallic acid. *L. plantarum* is equipped with the enzyme gallate decarboxylase, which decarboxylates gallic acid to pyrogallol [54,55]. With gallic acid being a food-derived metabolite, a potential dietary bias in the PROVIT cohort cannot be ruled out. Although all patients received the same hospital food during their stay, they were allowed to eat ad libitum.

Methylamine, synthesized from food-derived choline and lecithin by bacteria in the gut, was significantly increased in individuals receiving probiotic treatment. In the microbiome, choline and lecithin are metabolized by bacteria to trimethylamine (TMA), dimethylamine (DMA), and monomethylamine (MMA) [55]. In accordance with the increased levels of methylamine in the probiotics, trimethylamine was elevated in the probiotic group of the PROVIT trials after four weeks of probiotic treatment, compared to the beginning of the study. Besides the primary bacterial metabolites SCFA, bacteria in the gut also elicit the ability to produce middle-chain fatty acids (MCFAs), which was recently discovered by Gregor et al. (2021) [56,57,58].

Nevertheless, still little is known about the effects of MCFAs on the gut-microbiome. Gregor et al. (2021) demonstrated that a high-fiber diet in mice led to increased production of SCFA and capric acid [59]. This aligns with the results obtained in the PROVIT metabolomics analysis. Butyrate, an SCFA, and capric acid, an MCFA, were elevated and strongly correlated with each other after 28 days of the probiotic intake compared to the placebo group. In the same study, Gregor et al. (2021) showed that bacterially produced capric acid had an impact on inflammatory parameters in the colon [58].

In the PROVIT study, four weeks of probiotic supplementation resulted in a not distinctively different metabolomic profile in serum. However, this does not limit the findings of the PROVIT trial, as microbial metabolites show various methods of action to influence the host. Bacterial metabolites can either be systemically absorbed into the bloodstream or act locally in the gut [60]. Furthermore, a bidirectional communication pathway via microbiota in the colon and the vagus nerve exists. For example, the afferent fibers of the vagus nerve can affect the gut via a cholinergic anti-inflammatory pathway, which potentially decreases peripheral inflammation and intestinal permeability [59]. Additionally, microbial metabolites act as endogenous ligands. Butyrate, for example, is an endogenous ligand for orphan G protein-coupled receptors (GPCRs). In addition, butyrate can influence gene expression by inhibiting histone deacetylases [61].

Changing the microbiome composition affects the metabolome, resulting in important therapeutic implications, as MDD was clearly associated with metabolomic changes in large meta-analyses [20]. After further investigations in the future, probiotics could be reevaluated for clinical use. As MDD is a multifactorial disorder, it must be treated as such. This includes an interdisciplinary regimen according to state-of-the-art guidelines, consisting of modern personalized psychopharmacotherapy, psychotherapy, and lifestyle changes (stress reduction, daily structure, sports, and a healthy diet) [4]. The addition of probiotics can be a further piece in the treatment puzzle of individuals with MDD. As one-third of depressive patients are treatment-resistant, additional therapy options and personalized medication strategies are crucial to treat those affected by MDD and start the era of ‘psychobiotics’ [62]. In the future, personalized probiotic products with specific personalized bacterial strains could be adapted to the individual’s bacteria diversity in the gut. A tailored probiotic approach could direct the microbiome in the gut to an individual’s favorable composition. We assume that probiotics may also have potential preventive effects, as they elicit mild anti-inflammatory potential, which potentially counteracts chronic mild inflammation processes. Nevertheless, further research is necessary.

## 5. Limitations

The current investigation used standard statistical methods for metabolomics analyses. Overall, the effects are moderate and likely no longer significant when applying other statistical methods such as ANCOVA or Bonferroni correction. However, statistical analysis with Student’s *t*-test strongly indicates that probiotics have an effect on the metabolome of stool. In addition, we expect that the intervention period was too short and the sample size of the PROVIT Metabolomics cohort was too small to see differences in the two-way ANOVA analysis. This is also in line with weak correlations of stool metabolite levels, HAMD, and HAMD differences in baseline and intervention samples, respectively (Appendix A). Future RCTs with larger sample sizes and a more extended intervention period are needed to validate our findings. Furthermore, due to the clinical setting of the study, we could not consider the effects of different psychopharmacological treatments, as the medication changed frequently during patients’ stay on the ward. All drugs were recorded at the start of the PROVIT study and are listed in Table 1, and no statistical difference between the placebo and probiotic groups was observed. It was impossible to statistically control for the effects of pharmacological treatment during and after the 4 week intervention period in the metabolomics analysis. It is also conceivable that the medication had a strong influence on the metabolome, and the effects of probiotic treatment were masked. Additional studies need to be conducted, not only with a higher number of participants, but also with non-premedicated patients. Yet, these patients are difficult to recruit, because almost all patients with MDD are already premedicated, as are almost all patients who have to be admitted to a psychiatric ward. Nevertheless, psychopharmacological medication is necessary for severely ill patients and is therefore ethically necessary as we only investigated probiotics as an add-on therapy. As previously reported by Reininghaus et al. (2020), the 16S rRNA microbiome analysis allows us to study the bacterial species present in the gut, but it does not provide any information about the existing bacterial metagenome. To further reveal the influence of probiotic treatment on the biotransformational capabilities present in the microbiome, shotgun metagenomic sequencing needs to be performed.

The PROVIT Metabolomics investigation is the first pilot study of probiotic RCT in individuals with MDD, which analyzes alterations in the metabolomic profiles of serum and stool. A major strength of the PROVIT study is that the patients were exposed to similar environmental stimuli, such as hospital food and physical activity. Moreover, the administration of the placebo/probiotic drink was ensured by physicians daily.

## 6. Conclusions

This work has yielded several promising new avenues of research on influencing the microbiome in the gut by probiotic intake in individuals with MDD. Four weeks of probiotic add-on therapy resulted in higher normalized concentrations of butyrate, sarcosine, and methylamine in the stool of MDD patients (*n* = 28) than in individuals receiving the placebo (*n* = 29). Furthermore, amino acids, such as isoleucine, valine, lysine, and alanine, were elevated, whereas gallic acid was significantly reduced in the stool metabolome of individuals receiving probiotic add-on therapy. To summarize, NMR-based metabolomics is a sensible tool to identify differences in the stool’s metabolomic pattern induced by probiotic supplementation. Probiotics are likely to be a promising treatment add-on, especially for the prevention of stress-lifestyle-related disorders such as burnout and depression and can be a beneficial supplement during periods of special stress such as the COVID-19 pandemic [63]. Nevertheless, the overall significance of the metabolites altered by the probiotic treatment needs to be validated in future studies.

## Figures and Tables

**Figure 1 metabolites-12-00770-f001:**
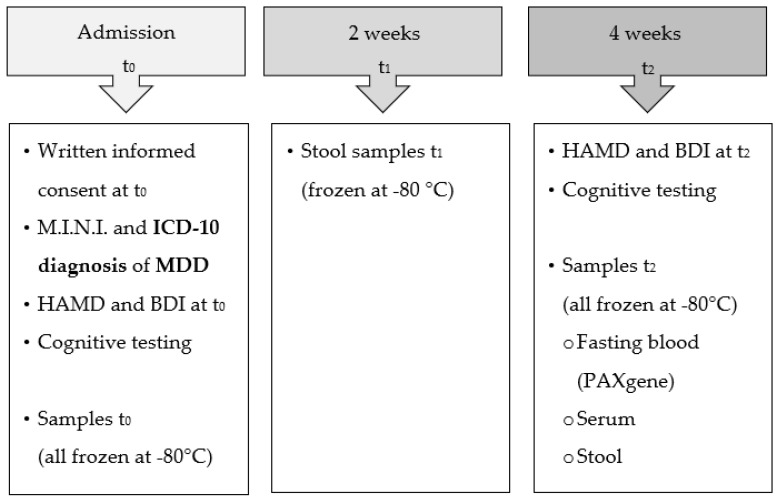
Overview of the PROVIT study design. HAMD = Hamilton Rating Scale for Depression, BDI-II = Beck Depression Inventory-II, M.I.N.I. = Mini International Neuropsychiatric Interview, ICD10 = International Statistical Classification of Diseases and Related Health Problems, MDD = Major Depressive Disorder.

**Figure 2 metabolites-12-00770-f002:**
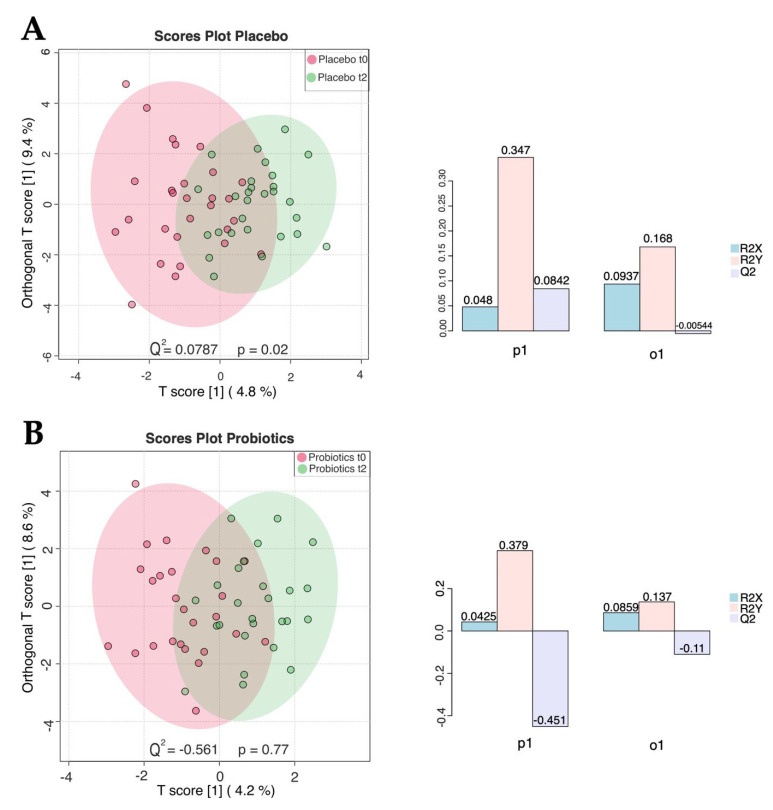
Untargeted metabolomics of the placebo (*n* = 29) and intervention group (*n* = 28) in serum. (**A**) OPLS-DA: T score of 4.8% and Orthogonal T score 2 of 9.4%. Showing minimal clustering Q^2^ of 0.0787 and *p*-value of 0.02. Time points: (1) Admission t_0_, (2) end of the trial t_2_ in the placebo group. (**B**) OPLS-DA: T score of 4.2% and Orthogonal T score 2 of 8.6%, showing no significant clustering Q^2^ of 0.0787 and *p*-value of 0.02. Time points: (1) Admission t_0_, (2) end of the trial t_2_ in the placebo group.

**Figure 3 metabolites-12-00770-f003:**
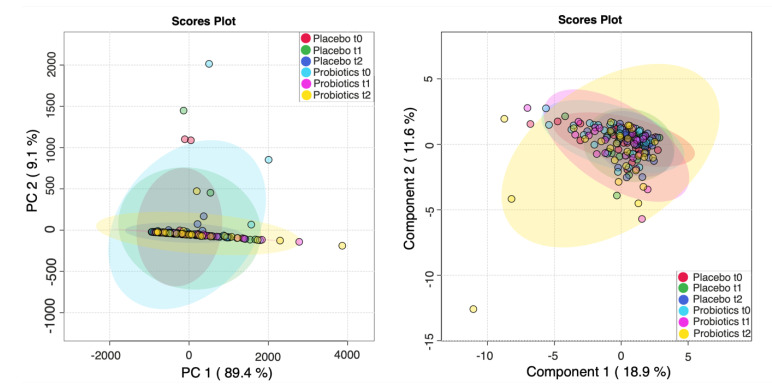
Untargeted metabolomics of the placebo (*n* = 29) and intervention groups (*n* = 28) in stool. (**A**) Principal component analysis (PCA), principal component (PC) 1 of 89.4% and PC2 of 9.1%. (**B**) Untargeted metabolomics of the placebo (*n* = 29) and intervention group (*n* = 28) in stool. Partial least-squares–discriminant analysis (sPLS-DA) Scores plot: Component 1 of 18.9% and Component 2 of 11.6%.

**Figure 4 metabolites-12-00770-f004:**
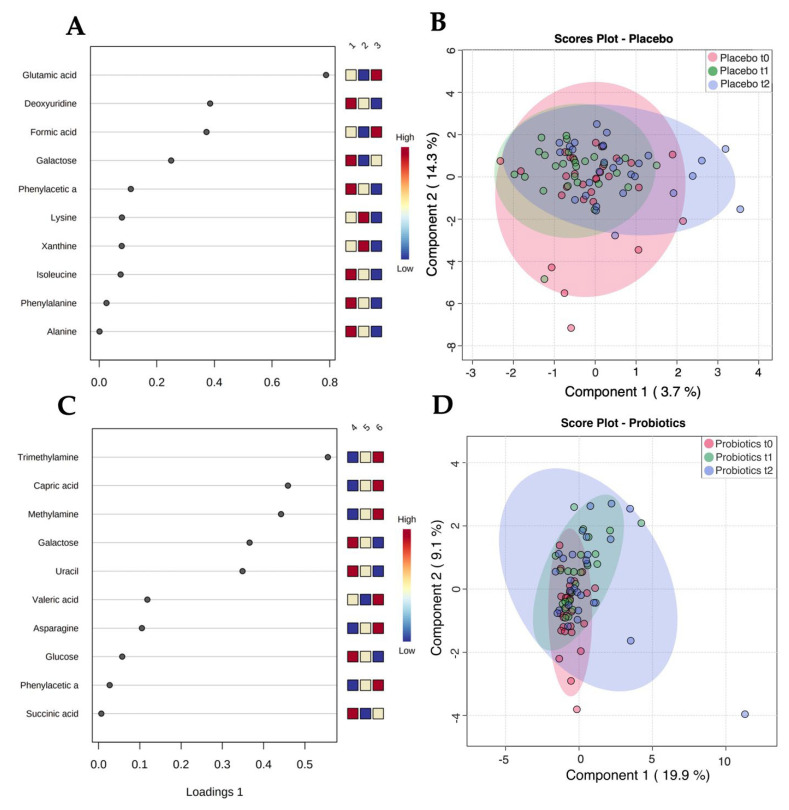
Untargeted metabolomics (**A**) of the placebo group (*n* = 29) in stool. Features indicating most distinctive metabolites. A heatmap is illustrated on the right side next to the table with the listed features, showing changes in concentration. Time points: (1) Admission t_0_, (2) two-week follow up t_1_, (3) end of the trial t_2_ in the placebo group (*n* = 28). (**B**) sPLS-DA scores plot of the placebo group (*n* = 29) in stool: Component 1 of 3.7% and Component 2 of 14.3%. (**C**) Untargeted metabolomics of the probiotics group (*n* = 28) in stool. Features indicating most distinctive metabolites. A heatmap is illustrated on the right side next to the table with the listed features, showing changes in concentration. Time points: (4) Admission t_0_, (5) two-week follow-up t_1_, (6) end of the trial t_2_ in the intervention group (*n* = 29). (**D**) sPLS-DA scores plot of the probiotic group (*n* = 28): Component 1 of 19.9%, Component 2 of 9.1%.

**Figure 5 metabolites-12-00770-f005:**
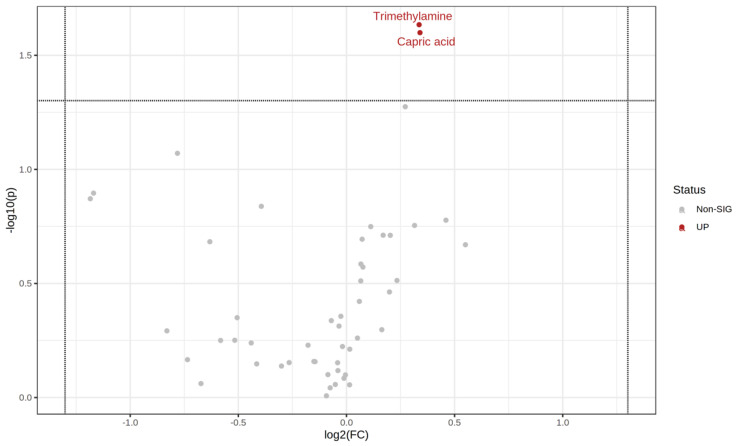
Volcano plot: Probiotics group longitudinal pairwise comparisons of time point t_0_ and t_2_ in stool. The log_2_ fold change (log2 FC) (*x*-axis) is plotted against the corresponding adjusted *p*-value (*y*-axis). Thresholds for significance (adjusted *p*-value = 0.05; horizontal dashed line) and changes in concentration, vertical dashed lines, are shown. Significantly increased metabolites in the probiotics group (*n* = 28) are displayed as red dots after 4 weeks compared to the beginning. Insignificantly altered metabolites are shown as gray dots.

**Figure 6 metabolites-12-00770-f006:**
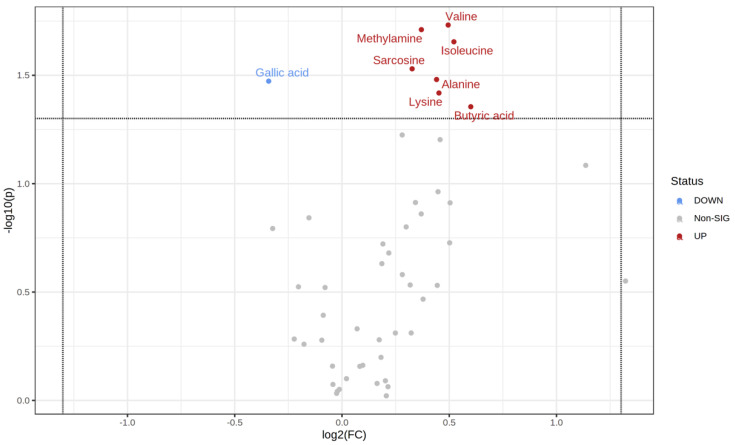
Volcano plot: Pairwise comparison placebo vs. probiotics time point t_2_ in stool. The log_2_ fold change (*x*-axis) is plotted against the corresponding adjusted *p*-value (*y*-axis). Thresholds for significance (adjusted *p*-value = 0.05; horizontal dashed line) and changes in concentration, vertical dashed lines, are shown. Significantly increased metabolites compared to probiotics at t_2_ are displayed as dots in dark red on the right side. Significantly decreased metabolites compared to probiotics (*n* = 28) are displayed as light blue dots on the left side. Upregulated features are displayed as red dots on the right side, whereas downregulated features are displayed as blue dots on the left side. Insignificantly altered metabolites are shown as gray dots.

**Figure 7 metabolites-12-00770-f007:**
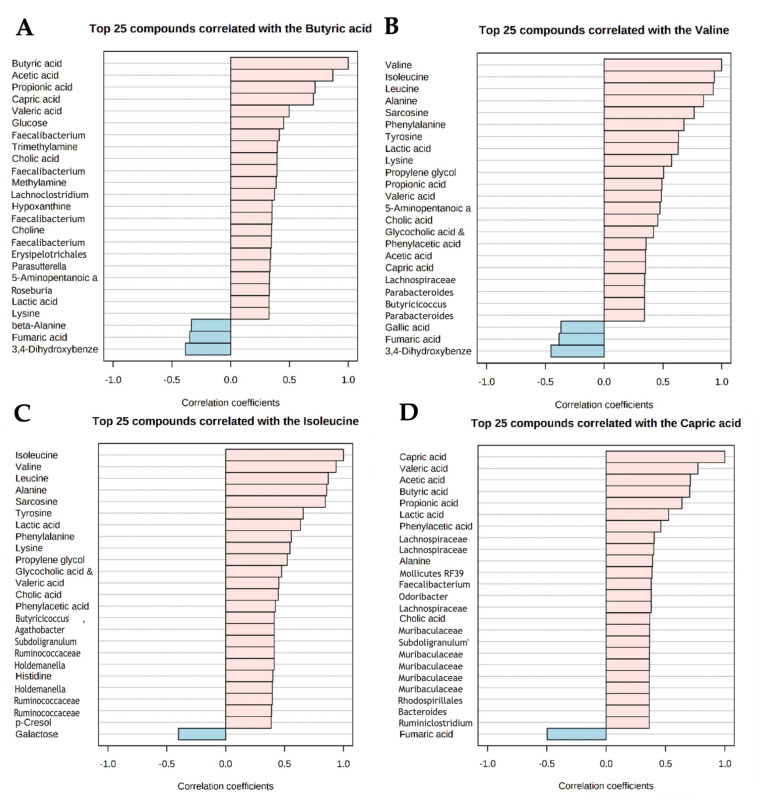
Top 25 correlations with the normalized concentrations of (**A**) valine, (**B**) butyric acid, (**C**) capric acid, (**D**) isoleucine, and 16S rRNA microbiome analysis data and the normalized concentrations of quantitatively assessed metabolites by NMR-Metabolomics.

**Table 1 metabolites-12-00770-t001:** Description of the PROVIT metabolomics sample at baseline.

Description	Intervention Group (N = 28)	Placebo Group (N = 29)	Statistics	
	N (%)	N (%)	*χ^2^*	*Sig (p)*
Sex (female)	21 (75.0%)	24 (82.8%)	0.516	0.473
Smoking (yes)	8 (29.6%)	16 (55.2%)	3.725	0.054
Dairy products before trial (yes)	13 (52.0%)	7 (28%)	3.000	0.086
	**Mean (SD)**	**Mean (SD)**	** *T* **	** *p* **
Age (years)	44.63 (15.12)	40.38 (11.30)	−1.205	0.233
Waist-to-hip-ratio	0.86 (0.07)	0.83 (0.10)	−0.969	0.337
	**Median** **(Mean rank)**	**Median** **(Mean rank)**	** *U* **	** *p* **
Education (years)	9.50 (28.02)	9.00 (28.95)	379	0.815
Illness duration (years)	6.00 (26.54)	11.00 (28.33)	339	0.677
BMI [kg/m^2^]	24.64 (27.52)	25.63 (30.43)	446	0.523
HbA1c (mmol/mol)	33.00 (25.31)	33.00 (26.61)	308	0.754

Note. Sig = Significance, SD = Standard Deviation, BMI = Body Mass Index, HbA1c = Haemoglobin.

**Table 2 metabolites-12-00770-t002:** Description of the pharmacological treatment of PROVIT metabolomics at baseline.

Description	Intervention Group (N = 28)	Placebo Group (N = 29)	Statistics	
	N (%)	N (%)	*χ^2^*	*Sig (p)*
Atypical antipsychotics	9 (32.1%)	10 (34.5%)	0.035	0.851
Anticonvulsants	3 (10.7%)	3 (10.3%)	0.002	0.964
Antihypertensive drugs	4 (14.3%)	1 (3.4%)	2.091	0.148
Benzodiazepines and hypnotics	5 (17.9%)	6 (20.7%)	0.073	0.786
Glutamatergic antidepressants	0 (0.0%)	1 (3.4%)	0.983	0.322
Melatonin-like antidepressants	0 (0.0%)	1 (3.4%)	0.948	0.330
Noradrenalin Dopamine Reuptake inhibitor	1 (3.6%)	3 (10.3%)	1.002	0.317
Noradrenergic and specific serotonergic antidepressants	1(3.6%)	2 (6.9%)	0.316	0.574
Selective serotonin reuptake inhibitor (SSRI)	11 (39.3 %)	13 (46.4 %)	0.292	0.589
Serotonin antagonist and reuptake inhibitor	15 (53.6%)	13 (44.8%)	0.436	0.509
Serotonin–norepinephrine reuptake inhibitors (SNRIs)	6 (21.4%)	11 (37.9%)	1.854	0.173
Proton pumps inhibitors	2 (7.1%)	4 (13.8%)	0.669	0.413
Tri- and tetracyclic antidepressants (TZA)	0 (0.0%)	2 (6.9%)	2.001	0.157
Thyroid medications	2 (7.1%)	5 (17.2%)	1.349	0.246

Note. Sig = Significance.

**Table 3 metabolites-12-00770-t003:** Mean values and standard deviation of depression scores of PROVIT metabolomics.

	Intervention Group (N = 28)		Placebo Group (N = 29)	
	Mean	SD	Mean	SD
BDI-II t_1_	31.11	8.43	33.39	10.14
BDI-II t_2_	16.00	8.86	19.28	11.13
HAMD t_1_	15.14	6.25	14.45	4.44
HAMD t_2_	9.30	5.61	8.28	5.91

Note: SD = Standard Deviation, BDI-II = Beck’s Depression Inventory, HAMD = Hamilton Depression Scale.

## Data Availability

The data presented in this study are available on request from the corresponding author. The data are not publicly available due to lack of a suited deposition platform.

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
