# Peer review of "The PROVIT Study—Effects of Multispecies Probiotic Add-on Treatment on Metabolomics in Major Depressive Disorder—A Randomized, Placebo-Controlled Trial"

_metabolites, 2022, doi:10.3390/metabo12080770_

Round 1

Reviewer 1 Report

It has long been debated the influence of the brain on the gut and vice-versa. We know that the gut-brain axis plays a role in major depressive disorder (MDD). Critical bacterial metabolites of the gastrointestinal tract are suspected to reduce low-grade inflammation and potentially influence brain function. Nevertheless, randomized, placebo-controlled probiotic intervention studies investigating metabolomic changes in patients with MDD have been scarce and presented no solid results. This work has delivered several promising new avenues of research on influencing the microbiome in the gut by probiotic intake in individuals with MDD. Although the overall significance of the metabolites altered by the probiotic treatment needs to be validated in future studies, it is promising that these results will influence major protocols in medicine in this decade.   The manuscript is well written, although some sentences occasionally seem awkward and can be revised. Occasional mistyping is also noted, Particularly for MDD, metabolomics of randomized, double-blind, placebo-controlled multispecies probiotic trials are still lacking, and the PROVIT  study is, to the next of my knowledge, the first study worldwide in MDD which analyzed microbiome, metabolome, gene expression, and cognition changes after a randomized, double-blind, placebo-controlled, multispecies probiotic intake with state-of-the-art molecular biological methods.

I congratulate the authors for their work. I found the manuscript an excellent piece of research. What is missing is the recruitment and emphasis of stakeholders that can be integrated in a paragraph in the discussion.

Author Response

Dear Reviewer 1,

we thank you very much for your precious time for reviewing the manuscript. Your suggestions improved the article to a high degree.

Question:

The manuscript is well written, although some sentences occasionally seem awkward and can be revised. Occasional mistyping is also noted.

Answer: Thank you very much for your suggestion. We proof-read the text carefully and corrected the typos.

Question:

I congratulate the authors for their work. I found the manuscript an excellent piece of research. What is missing is the recruitment and emphasis of stakeholders that can be integrated in a paragraph in the discussion.

Answer: We added the following sentences highlighted in green based on your suggestions in the methods section:

„The participants were recruited at the Clinical Department of Psychiatry and Psychotherapeutic Medicine at the Medical University of Graz by psychiatrists and psychologists of the research unit for affective disorders managed by Univ.Prof. Eva Reininghaus.“

2.3. Details of the probiotic supplementation

Individuals subjected to the verum group received a multistrain probiotic ‘OMNi-BiOTiC® Stress Repair’, which was provided by the ‘Institute Allergosan’ and produced by Winclove BV, Amsterdam, Netherlands.

“The drink was provided by a psychiatrist on call from the research unit every day in the morning from 7 to 8 o’clock before breakfast. The doctors on call were in charge to stir the drink in a double- blind-setting and surveyed the intake of the probiotic/placebo drink.“

We also added the recruitment process and the participation of Allergosan in the Discussion

The current PROVIT investigation, which was recruited at the Department of Psychiatry and Psychotherapeutic Medicine at the Medical University of Graz in Austria, showed a significant shift in the stool’s metabolomic profile of depressed individuals receiving 28 days of multispecies probiotic treatment that was provided by Allergosan. 

Reviewer 2 Report

The Manuscript investigates the effect of probiotic blend supplementation in subjects affected by major depressive disorder from a metabolomic and microbiological point of view. A microbiota-based therapy, together with the pharmacological treatment of patients, is a promising future application for depression.

Title

Put a dash between the words “placebo” and “controlled” (page 1, line 4).

Keywords

What is “RCT”? Please specify (page 1, line 48).

Introduction

Delete the comma (page 2, line 52).

Put a comma after the word “deficits” (page 2, line 56).

Figure 1

It does not seem like a figure but rather a table. The Author may consider changing the format in which the information is presented. Please specify the acronyms used “M.I.N.I”, “ICD-10”, “HAMD”, “BDI”.

2.3. Metabolomics and 2.5. Statistics

Delete the extra dot in both cases.

Table 1. Description of the pharmacological treatment of PROVIT metabolomics at baseline

It should be named Table 2, as referred in the text. Please check for the consequential number of tables.

Table 2. Mean values and standard deviation of depression scores of PROVIT metabolomics

No refence to Table 2 is present in the text.

Results

Lack of references in the text to many figures belonging to the Supplementary file. Please describe all the results in this section.

Discussion

Put the name of the bacterium in italicum (page 14, line 393).

All along the Manuscript, check for extra spaces and put a space between a number and its unit. Correct the spelling of “fours” (page 12, line 365).

Did the Author find an individual metabolic signature (metabotype), stable over time, allowing them to monitor individual status and response to different stimuli in their work?

Author Response

Corrections requested by the reviewer

Dear Reviewer 2,

thank you very much for your suggestions, which improved the manuscript to a high degree. We changed the following suggestions in the manuscript:

Title - Put a dash between the words “placebo” and “controlled” (page 1, line 4).

Keywords - What is “RCT”? Please specify (page 1, line 48).

Introduction - Delete the comma (page 2, line 52).

Put a comma after the word “deficits” (page 2, line 56).

 Question:

Figure 1. It does not seem like a figure but rather a table.

Answer: We changed the caption from Fig. 1 to Table 1 as you suggested.

Question:

The Author may consider changing the format in which the information is presented. Please specify the acronyms used “M.I.N.I”, “ICD-10”, “HAMD”, “BDI”.

Answer: Thank you for this precious point- we specified the mentioned acronyms and highlighted in green in the manuscript.

Question:

2.3. Metabolomics and 2.5. Statistics - Delete the extra dot in both cases.

Answer: We deleted it.

Question:

Table 1. Description of the pharmacological treatment of PROVIT metabolomics at baseline

It should be named Table 2, as referred in the text. Please check for the consequential number of tables.

Anwer: We named it Table 2 and checked the consequential number.

Question:

Table 2. Mean values and standard deviation of depression scores of PROVIT metabolomics

No refence to Table 2 is present in the text.

Results - Lack of references in the text to many figures belonging to the Supplementary file. Please describe all the results in this section.

Answer: We referred to the Table in the text as you suggested.

Question:

Discussion - Put the name of the bacterium in italicum (page 14, line 393).

Answer: We changed the bacterium in italicum

Question:

All along the Manuscript, check for extra spaces and put a space between a number and its unit. Correct the spelling of “fours” (page 12, line 365).

Answer: Thank you very much for your suggestion. We proof-read the text carefully and corrected the typos.

Did the Author find an individual metabolic signature (metabotype), stable over time, allowing them to monitor individual status and response to different stimuli in their work?

We saw that the metabolomic phenotype of the probiotic group became more heterogeneous in comparison to the placebo group. We identified metabolites, which differed significantly between placebo and probiotic group. Nevertheless, we could not identify individual patterns yet as we analyzed them on group level. Thank you for the recommendation for future studies.

Reviewer 3 Report

Review of The PROVIT study – Effects of multispecies probiotic add-on treatment on metabolomics in major depressive disorder – a randomized, placebo controlled trial (metabolites-1857730)

This study investigated the effect of multispecies probiotic on metabolites on the patients with major depressive disorder and showed that the stool metabolites of patients with major depressive disorder were changed by usage multispecies probiotic. The content of this study was interesting; however, several problems to be solved.

1.     In this study, the authors showed that the use of multiple probiotics changed only the metabolites in the stool. What effect can be expected only metabolites in stools were changed without serum metabolites change?

2.     Please add the analysis of the association between change of stool metabolites and change of the symptoms of major depressive disorder. This reviewer thinks that the data of (stool) metabolites, which were associated with change of the symptoms of major depressive disorder, are worth presenting in this journal.

Author Response

Dear Reviewer 3,

Thank you very much for using your precious time for reviewing. Your suggestions improved the manuscript a lot. We changed the manuscript based on your suggestions:

Question:

The content of this study was interesting; however, several problems to be solved.

  1. In this study, the authors showed that the use of multiple probiotics changed only the metabolites in the stool. What effect can be expected only metabolites in stools were changed without serum metabolites change?

Answer:

Summarizing, we expect that the time frame was too short in the PROVIT sample and that the changes in the microbiome would be reflected in a change in the serum metabolome in a further study with longer time spans.

Butyrate, a metabolite which was significantly altered in the probiotic group, has an important impact on the permeability of intestinal epithelial cells (IEC). Butyrate strengthens the intercellular integrity and nourishes colonocytes. Thus, butyrate is an important player in immune functioning in the intestinal mucosa (Bach et al. 2018).

The immune system and the gastrointestinal tract are closely interconnected. Butyrate for example, promotes dendritic cells (DCs) to express enzymes, such as indoleamine 2,3 dioxygenase (IDO1) and aldehyde dehydrogenase 1A2 (Aldh1A2). IDO1 and Aldh1A2 can suppress the conversion of naïve T-cells into pro-inflammatory interferon producing cells. This mechanism happens via converting naive T-cells into immunosuppressive forkhead box P3(+) (FoxP3(+)) Tregs (regulatory T-cells) (Gurav et al. 2015). As chronic low-grade inflammation is also involved in the pathogenesis of depression, we expect that a favourable diversity of microbial metabolites (such as elevated levels of butyrate), result in a better gut health.

Most recently, a further probiotic intervention study investigating the metabolomics changes revealed an interesting influence of butyrate on the metabolome of type 2 diabetes patients. McMurdie et al. 2021 published a 12-weeks probiotic intervention trial with the bacterial strain WBF-011 in twenty-one T2D patients versus sixteen participants taking placebo. As a result, they could show a clear tendency between butyrate levels in stool and plasma with HbA1c. Further, an increase in butyrate and ursodeoxycholate and corresponding butyrate levels in stool and plasma were reported in McMurdie et al. 2021.  Thus, we expect that the changes in the microbiome would be reflected in a change in the serum metabolome in a further study with longer time spans.

References:

Bach Knudsen KE, Laerke HN, Hedemann MS, Nielsen TS, Ingerslev AK, Gundelund Nielsen DS, Theil PK, Purup S, Hald S, Schioldan AG, Marco ML, Gregersen S, Hermansen K: Impact of Diet-Modulated Butyrate Production on Intestinal Barrier Function and Inflammation. Nutrients 2018, 10(10):10.3390/nu10101499.

Gurav A, Sivaprakasam S, Bhutia YD, Boettger T, Singh N, Ganapathy V: Slc5a8, a Na+-coupled high-affinity transporter for short-chain fatty acids, is a conditional tumour suppressor in colon that protects against colitis and colon cancer under low-fibre dietary conditions. Biochem J 2015, 469(2):267-278.

  1. McMurdie PJ, Stoeva MK, Justice N, Nemchek M, Sieber CMK, Tyagi S, Gines J, Skennerton CT, Souza M, Kolterman O, Eid J: Increased circulating butyrate and ursodeoxycholate during probiotic intervention in humans with type 2 diabetes. BMC Microbiol 2022, 22(1):19-021-02415-8.
  2. Please add the analysis of the association between change of stool metabolites and change of the symptoms of major depressive disorder. This reviewer thinks that the data of (stool) metabolites, which were associated with change of the symptoms of major depressive disorder, are worth presenting in this journal.

Thank you for your suggestions. We analyzed the correlations of HAMD and BDI with stool metabolites. Although HAMD and BDI-II show the strongest correlations, the correlations between metabolites and HAMD and differences in HAMD between baseline and intervention samples, respectively, were weak. We added a summarizing statement to the revised version of the manuscript and the figures to the supplementary materials.

Figure S11. Top 25 Correlations coefficients of metabolites (normalized concentrations) with the HAMD.

Figure S12. Top 25 Correlations coefficients of metabolites (normalized concentrations) with the HAMD difference between baseline and intervention samples.

Round 2

Reviewer 3 Report

The content itself was summarized well. However, the main result of this study was that the probiotics change the metabolites in stool through changing microbiota. This reviwer thinks that this study lacks novelty.